# A Stride Toward Wine Yield Estimation from Images: Metrological Validation of Grape Berry Number, Radius, and Volume Estimation

**DOI:** 10.3390/s24227305

**Published:** 2024-11-15

**Authors:** Bernardo Lanza, Davide Botturi, Alessandro Gnutti, Matteo Lancini, Cristina Nuzzi, Simone Pasinetti

**Affiliations:** 1Department of Mechanical and Industrial Engineering (DIMI), University of Brescia, Via Branze 38, 25123 Brescia, Italy; bernardo.lanza@unibs.it (B.L.); davide.botturi@unibs.it (D.B.); cristina.nuzzi@unibs.it (C.N.); 2Department of Information Engineering (DII), University of Brescia, Via Branze 38, 25123 Brescia, Italy; alessandro.gnutti@unibs.it; 3Department of Medical and Surgical Specialties, Radiological Sciences, and Public Health (DSMC), University of Brescia, Viale Europa 11, 25123 Brescia, Italy; matteo.lancini@unibs.it

**Keywords:** measurement science, machine vision, viticulture, deep learning, fruit counting, fruit size estimation

## Abstract

Yield estimation is a key point theme for precision agriculture, especially for small fruits and in-field scenarios. This paper focuses on the metrological validation of a novel deep-learning model that robustly estimates both the number and the radii of grape berries in vineyards using color images, allowing the computation of the visible (and total) volume of grape clusters, which is necessary to reach the ultimate goal of estimating yield production. The proposed algorithm is validated by analyzing its performance on a custom dataset. The number of berries, their mean radius, and the grape cluster volume are converted to millimeters and compared to reference values obtained through manual measurements. The validation experiment also analyzes the uncertainties of the parameters. Results show that the algorithm can reliably estimate the number (MPE=−5%, σ=6%) and the radius of the visible portion of the grape clusters (MPE=0.8%, σ=7%). Instead, the volume estimated in ${\mathrm{px}}^{3}$ results in a MPE=0.4% with σ=21%, thus the corresponding volume in ${\mathrm{mm}}^{3}$ is affected by high uncertainty. This analysis highlighted that half of the total uncertainty on the volume is due to the camera–object distance *d* and parameter *R* used to take into account the proportion of visible grapes with respect to the total grapes in the grape cluster. This issue is mostly due to the absence of a reliable depth measure between the camera and the grapes, which could be overcome by using depth sensors in combination with color images. Despite being preliminary, the results prove that the model and the metrological analysis are a remarkable advancement toward a reliable approach for directly estimating yield from 2D pictures in the field.

## 1. Introduction

Grapes represent one of the world’s most prized crops, commanding a significant presence in a rising market encompassing diverse products such as fresh table fruit, raisins, wine, distillates, and juice concentrate. These commodities emerge from various cultivar species, predominantly categorized as white, red, and black varieties. The size and form of these berries exhibit notable variation depending on the grape species [1].

The estimation of the yield volume, which is the main objective of the present work, is a fundamental theme for farmers regardless of the specific cultivar they grow, allowing for efficient crop field management [2,3]. In viticulture, winegrowers and agronomists rely on manual measurements to gauge field yield, encompassing vine count, grape clusters per vine, and berries per cluster. These factors converge to estimate total harvested fruit weight and overall field productivity [4,5]. In fruit production, a good way to automatize this process requires obtaining an accurate estimation of the total number of fruits (yield counting task). This is the primary reason why several works nowadays focus on fruit counting. Finally, to obtain the final yield weight, the volume of the fruits (and ultimately their density) should be estimated as well (volume estimation).

A potential solution to both tasks is to use vision systems. Recent developments in computer vision (CV) and artificial intelligence (AI) research have paved the way for a remarkable array of applications that were inconceivable just a decade ago. These applications harness a range of 2D vision systems, ranging from color cameras to sensors operating beyond the visible spectrum, such as near-infrared (NIR), thermal, and hyperspectral cameras. Three-dimensional vision systems offer an additional option, allowing for a scene representation as a set of points in three dimensions. Nevertheless, when used in the field, surface reflections due to water droplets and ambient light interfere significantly with the quality of the acquired data. Field-based 3D reconstruction lacks the necessary accuracy and spatial resolution to be a trustworthy resource for measuring tiny fruits like grape berries. Moreover, the resolution of 3D cameras is typically lower than that of 2D cameras. This is important in situations (like counting grape berries) where the measured object has decreased dimensions [6]. As a result, despite being a promising technology given its extensive adoption in controlled environments, it is still not feasible to use 3D cameras for open-field acquisitions if the goal is the accuracy and robustness of the measurement of small fruits.

Among the works available in the literature, the majority deal with the task of berry counting while only a few deal with yield volume estimation. Since fruits can be approximated with regular geometrical shapes, it is sufficient to acquire a few key measurements from which the complete 3D shape can be derived. As a result, the research community has predominantly directed attention toward 2D image analysis. This choice was also motivated by the rapid development and widespread adoption of Deep Learning (DL) [7] techniques to analyze complex data and 2D images. Among the plethora of algorithms, Neural Networks (NNs) are the choice preferred by many, and some works even tried to adapt such models to agricultural tasks to automatically segment grape berries from color images [8]. Other studies have further explored the capability of DL models by experimenting with input parameters like the image color space (e.g., RGB, HSV, CieLAB, YUV), model architecture, and the influence of diverse augmentation techniques [9,10]. These studies are crucial in the development and design of robust algorithms and NNs capable of generalizing across grape varieties that differ not only in color (ranging from a light green coloring to a dark red accordingly) but also in the berry shape and size (ranging from almost spherical to an elongated ellipsoid). An encouraging method for counting individual berries within clusters was showcased in [11], where Luo et al. introduced a custom algorithm leveraging the berry’s edge contour, concave points, and curvature for accurate counting. This idea of utilizing the berry’s edge to enhance berry segmentation was further expanded in [12], in which Zabawa et al. delineate the berry’s edge as a new segmentation category, alongside whole berries and the background.

Despite substantial progress in fruit counting methodologies, especially those utilizing AI and CV, the literature indicates a considerable deficiency in strategies that can concurrently estimate geometric attributes like berry radius and volume directly from color images. Although many models have proven effective at counting individual fruits, they typically conclude with object detection and segmentation. Thus, the geometric characterization of each berry (i.e., measure of radius and volume of berries) is not investigated.

In light of the above-mentioned challenges, the present study details the novel weakly supervised neural network named STEWIE we proposed in our previous publication [13]. This model leverages a novel approach for simultaneously estimating both (i) the total number of berries within an image and (ii) their average radius. This innovative technique encompasses the use of a customized NN that generates density maps to predict the number of berries in each cluster and their average radius in pixels. To the best of our knowledge, there are no works that attempt to output the estimation of a geometrical feature (e.g., the radius) of a fruit directly from AI models. The only two works that tried to achieve this goal adopted a traditional approach leveraging CV and image processing techniques after the fruit segmentation phase performed by an AI model [11,12]. The novel contribution of the present article is the metrological validation of STEWIE [13], carried out by defining an experimental set-up and by evaluating STEWIE’s capabilities in predicting the volume of grape bunches, which is the ultimate goal of yield prediction. A thorough investigation of visible detected volume and the corresponding associated uncertainty is presented, a topic typically underestimated by the research community but fundamental to assess the performances of measurement systems.

## 2. Materials and Methods

### 2.1. Materials

This study aims to conduct a metrological validation of the STEWIE model presented in [13], evaluating its accuracy and reliability in estimating grape yield parameters in real-world situations, thus confirming its practical relevance in viticulture. STEWIE’s neural network takes an input image with dimensions H×W and produces two density maps, $D^{n}$ and $D^{r}$, both of size H×W. These density maps are employed to predict both the estimated number of berries ($\tilde{N}$) and their estimated average radius (${\tilde{r}}_{\mathrm{mean}}$), respectively (refer to Figure 1). The reader is encouraged to read the corresponding literature for technical details about the network structure, ground truth definition, and model training.

### 2.2. Used Datasets

In this work, we used two image datasets: (i) a validation dataset for the algorithm training as in [13] and (ii) a test dataset for the metrological validation of the approach.

The validation dataset adopted was the Embrapa Wine Grape Instance Segmentation Dataset (WGISD) [14], which includes 300 images of grape clusters from five different grape varieties (Chardonnay, Cabernet Franc, Cabernet Sauvignon, Sauvignon Blanc, and Syrah), with variations in pose, illumination, and focus, as well as genetic and phenological differences. As reported in the corresponding dataset article [14], an EOS REBEL T3i DSLR camera (Canon Inc., Tokyo, Japan) and a Z2 Play smartphone (Motorola Inc., Schaumburg, IL, USA) were used to capture the images. The cameras were positioned between the vine lines at 1 to 2 m, with the EOS REBEL T3i camera capturing 240 images, including all Syrah pictures, and the Z2 Play smartphone taking 60 images of all other grape varieties. The resulting images were scaled to 2048×1365 pixels for the EOS REBEL T3i DSLR and 2048×1536 pixels for the Z2 Play. Additional details about the image capture process can be found in the Exif data of the original image files, which are included in the dataset. In all 300 images, Geng Deng et al. [15] provided dot annotations identifying a total of 187,374 berries. Image examples taken from the dataset are shown in Figure 2.

To evaluate the performance of STEWIE [13] and estimate measurement uncertainty, a dedicated test dataset was specifically created in this work. The chosen grape variety for this evaluation was the Flame variety characterized by red and round berries. This variety was purposely chosen to assess the model’s generalization capacity because it was not included in the original training dataset. The test dataset included B=10 red grape clusters, from which 3 images were captured per cluster, leading to a cumulative set of K=30 images. These images were acquired within a controlled environment using the low-cost camera Arducam AR0234 (Arducam, Nanjing, China) while being exposed to outdoor conditions to factor in natural lighting and real-world background irregularities. To capture the set of 3 images for each grape bunch, we followed a process where we individually suspended each bunch on a vine tree positioned outdoors, maintaining a fixed distance of d=500 mm from the camera. The initial image was taken in this configuration, while the subsequent two images were captured after rotating the bunch by $120^{\circ }$ and $240^{\circ }$, respectively, around its vertical central axis. This approach allows to account for orientation variability in our analysis. This variability could either enhance or impede the performance of the image analysis software, depending on factors such as the occlusion of certain berries and the presence of illumination noise.

### 2.3. Camera Calibration

Since the volume estimation should be provided in metric units to winegrowers, a calibration procedure must be conducted on the involved 2D cameras to estimate the intrinsic camera matrix needed to convert pixel data to millimeters [16,17]. The matrix contains the coordinates of the optical center ($c_{x},c_{y}$) and the focal length *f* of the camera. The procedure was conducted using MATLAB computer vision toolbox (MathWorks Inc., Natick, MA, USA) [18,19]. Considering the set-up described in Section 2.2, we captured 30 images of a checkerboard pattern with squares of 20 mm each, glued on a rigid and planar support. The images were taken at different distances and orientations to improve the estimation result of the intrinsic matrix. To convert pixel values to the corresponding ones in millimeters, Equation (Equation 1) was applied (the object-camera distance *d* was set to 500 mm in our experiments).
(1)$$C_{px-mm}=\frac{d}{f}$$

### 2.4. Model Validation

To assess the efficacy of the model, it is necessary to (i) quantify the real number of berries within each grape bunch and (ii) obtain an estimated measurement of their effective radii. The Flame variety used in the test dataset is known for its round berries; hence, we assumed that the shape of the berries could be approximated as a sphere. To verify this assumption, we manually measured the berries of the bunches in the dataset using a caliber with a resolution of 0.01 mm. This ensured that the collected diameters did not exhibit significant differences, confirming the validity of the spherical model. Thus, manual annotation was performed on each image, enabling the acquisition of (i) the count of observable berries that STEWIE aimed to identify and (ii) the corresponding radii associated with these berries. Figure 3 visually presents examples of the labeled data. A summarized representation of the manual measurement data associated with each bunch is presented in Table 1. This table includes (i) the unique bunch ID number, (ii) the total number of berries within the bunch ($N_{T}$), (iii) the count of visible berries in each image for the respective bunch ($N_{i}$, with i=1, …, 3, where i=1 represents the image taken in standard configuration, i=2 the image taken after rotating the bunch of $120^{\circ }$, and i=3 the image taken after rotating it of $240^{\circ }$), (iv) the mean radius of berries (with their standard deviation) within the entire bunch in millimeters ($r_{\mathrm{mean},T}\pm \sigma$), and (v) the average radius of the berries in pixels, computed manually based on the visible berries within each image ($r_{\mathrm{mean},i}$).

#### 2.4.1. Visible Berry Counting Validation

For the validation of the visible berry counting task, the model outputs obtained from the K=30 test images must be compared with the real measured information summarized in Table 1. To achieve this, metrics such as the mean error (ME) and the mean percentage error (MPE) between the actual number of berries and the estimated number of berries within the test images were used (Equation (Equation 2)).
(2)$$\begin{array}{cc} \mathrm{ME} & =\frac{1}{K}\sum_{k=1}^{K}{\tilde{N}}_{k}-N_{k}\\ \mathrm{MPE} & =\frac{1}{K}\sum_{k=1}^{K}\frac{{\tilde{N}}_{k}-N_{k}}{N_{k}} \end{array}$$

In Equation (Equation 2), ${\tilde{N}}_{k}$ and $N_{k}$ represent the number of berries obtained from the algorithm and the real number of berries, respectively.

As further validation information, the berry counting output obtained on the test dataset should be compared with those acquired from the validation dataset (see Section 2.2). However, images in the WGISD dataset were captured in a field setting, often containing multiple clusters within a single image. As a result, the algorithm’s estimation was applied to the image contents as a whole, rather than individual bunches. On the other hand, images in the test dataset were taken in a controlled set-up, each depicting a single cluster of a grape variety not present in the original training and validation datasets (see reference [13] for details about model training).

As a result, it is necessary to extrapolate the outcomes from the single-cluster-per-image scenario to the context of multiple clusters. To achieve this, we empirically verified that the prediction error of STEWIE on test images portraying a single bunch conforms to a normal distribution with a mean equal to ME and a standard deviation equal to $\sigma_{E}$. Thus, we designated the probability density function of the error for these single-bunch images as $f_{E}\left(x\right)=N\left(\mathrm{ME},\;\sigma_{\mathrm{ME}}^{2}\right)$. If we analyzed an image with B>1 bunches, the task would have involved computing the probability density function $f_{E_{B}}\left(x\right)$ for errors in images containing *B* bunches. As a result, $f_{E_{B}}\left(x\right)$ is the probability density function associated with the summation of *B* normal variables. Hence, the expression is reported in Equation (Equation 3) (the formula is valid under the fair assumption of independence between the errors for the bunches within the same image).
(3)$$f_{E_{B}}\left(x\right)=N\left(B\cdot \mathrm{ME},\;B\cdot \sigma_{E}^{2}\right)$$

Therefore, it is possible to estimate ME^*B*^, MAE^*B*^, and RMSE^*B*^ (root mean squared error) for an image containing *B* clusters by using Equation (Equation 4).
(4)$$\begin{array}{cc} {\mathrm{ME}}^{B} & =\int_{-\infty }^{+\infty }xf_{E_{B}}\left(x\right)\;dx\\ {\mathrm{MAE}}^{B} & =\int_{-\infty }^{+\infty }\left|x\right|f_{E_{B}}\left(x\right)\;dx\\ {\mathrm{RMSE}}^{B} & =\sqrt{\int_{-\infty }^{+\infty }x^{2}f_{E_{B}}\left(x\right)\;dx} \end{array}$$

As the quantity of clusters within the validation images is not constant, we considered the average number B=14 (this value was obtained by manually analyzing the images contained in the validation dataset).

#### 2.4.2. Berry Radius Estimation Validation

To validate the model’s capacity for accurate berry size estimation, the radius estimation results obtained on the test images are compared against the manually measured radius values (ground truth) listed in Table 1. The difference between the ground truth and the estimation constitutes the model’s estimation error, computed using Equation (Equation 2). For this computation, the annotated mean radius $r_{\mathrm{mean},k}$ and STEWIE’s estimation of the average radius ${\tilde{r}}_{\mathrm{mean},k}$ are used in place of $N_{k}$ and ${\tilde{N}}_{k}$, respectively.

#### 2.4.3. Volume Estimation Validation

As a final contribution, we derived the grape volume by using the estimated quantities of the number of berries and their average radius in pixels. This volume estimation can subsequently serve as a basis for farmers and wine producers to accurately calculate the yield [3,4].

To validate the volume estimation, we need the effective volume of each bunch of the test dataset. The validation was performed considering (i) the volume of the entire bunch $V_{b,mm}$ and (ii) the volume of the visible part of the bunch $V_{I,px}$ (because 2D images depict only a portion of the overall berries in the bunch due to occlusions). The ground truth values for the volumes of the bunches in ${\mathrm{mm}}^{3}$ were obtained by manually measuring the diameter of each grape berry of each bunch with a caliber (as described in Section 2.4), while the reference values for the computation of the volume of visible part of the bunch (in ${\mathrm{px}}^{3}$) were obtained by manually annotating each grape berry on the images together with their diameters. Both volumes were derived considering the hypothesis of spherical berries.

Thus, we first computed the volume ${\tilde{V}}_{I,px}$ of the visible part of the bunch and compared it with the corresponding visible volume. From ${\tilde{V}}_{I,px}$, we then derived the volume of the entire bunch, ${\tilde{V}}_{b,mm}$.

To facilitate the subsequent discussion, we clarify the notation as follows:$r_{mm}$ is the radius in metric coordinates of a berry in the bunch that was manually measured using a caliber;$r_{px}$ is the radius in pixels of a berry present in an image that was manually measured from the image;$V_{I,px}$ represents the volume of the bunch *b* in $px^{3}$, considering only the berries visible in the image. This is approximated as $\sum_{n=1}^{N}\frac{4}{3}\pi r_{px,n}^{3}$;$V_{I,mm}$ represents the volume of the bunch *b* in $mm^{3}$, considering only the berries visible in the image. This is approximated as $\sum_{n=1}^{N}\frac{4}{3}\pi r_{mm,n}^{3}$, where N<M represents the number of berries in the image;$V_{b,mm}$ defines the volume of the bunch *b* in $mm^{3}$. It is approximated as $\sum_{m=1}^{M}\frac{4}{3}\pi r_{mm,m}^{3}$. Here, *M* represents the total number of berries in the bunch, and $r_{mm,m}$ is the radius of the *m*th berry.

The estimated volume is computed using Equation (Equation 5), in which ${\tilde{N}}_{k}$ denotes the estimated number of berries and ${\tilde{r}}_{\mathrm{mean}}$ is the mean radius estimated by STEWIE.
(5)$${\tilde{V}}_{I,px}={\tilde{N}}_{k}\frac{4}{3}\pi {\tilde{r}}_{\mathrm{mean}}^{3}$$

To evaluate the accuracy of this estimation, we calculated ME and MPE (in pixels) between nominal volume $V_{I,px}$ and volume estimated from images ${\tilde{V}}_{I,px}$ (see Equation (Equation 2) for math computation). These metrics were derived by averaging the errors observed in individual images within the test dataset.

In a practical agricultural context, farmers are interested in obtaining a rough estimate of the total weight of their grape yield. To achieve this, the goal is to extend the estimated visible volume ${\tilde{V}}_{I,px}$ to estimate the volume of the entire grape cluster. Therefore, we need to determine ${\tilde{V}}_{b,mm}$, which represents the estimated volume of the entire grape cluster in metric units. To accomplish this, two steps are necessary: (i) convert ${\tilde{V}}_{I,px}$ to metric units ${\tilde{V}}_{I,mm}$ using the conversion factor $C_{px-mm}$ that converts pixel units to millimeters (estimated by the camera calibration procedure as described in Section 2.3) and (ii) multiply the result by factor *R*, which takes into account the proportion of visible grapes with respect to the total grapes within cluster *b* in image *k*.

In this initial investigation, we proposed to calculate parameter *R* as the ratio between the volume of the entire grape bunch $V_{b,mm}$ and the volume of the visible grapes $V_{I,mm}$, averaged across all the images in our test dataset, which consists of 30 images. This parameter is crucial in our validation experiment because we measured the entire bunch volume manually; however, only a portion can be seen in the images since some berries were be obscured by the foreground ones, thus leading to an underestimation of the visible volume in the images. Hence, parameter *R* was used as a correction factor. It is worth noting that we introduced some bias into the calculation by utilizing the ground truth information from the dataset. Ideally, we would select a small subset of grape bunches to calculate the conversion factor *R* and then validate its applicability on the remaining dataset. However, due to the constraints of our limited dataset, we chose the approach described above, which remains a reasonable and practical solution for this preliminary analysis. As a result, the estimated volume of the whole grape bunch in metric units is computed by using Equation (Equation 6).
(6)$${\tilde{V}}_{b,mm}={\tilde{V}}_{I,px}\cdot C_{px-mm}^{3}\cdot R$$

This information serves as the final output for farmers, aiding them in estimating the total yield of their grape harvest. It is important to note that this value is subject to uncertainty.

### 2.5. Uncertainty Evaluation for Volume Estimation

Since the final output for the farmers is ${\tilde{V}}_{b,mm}$, it is necessary to evaluate the uncertainty of each variable that contributes to its calculus by using the “Guide to the Expression of Uncertainty” (GUM, linear propagation, simplified approach) [20,21].

Before estimating the uncertainty of ${\tilde{V}}_{b,mm}$, we first need to estimate the uncertainty of $V_{I,px}$ (obtained from Equation (Equation 7)) and $V_{b,mm}$ (obtained from Equation (Equation 8)) to ensure that they can be taken as reference values. To achieve this, we need to evaluate the uncertainty of (i) $r_{px}$, and (ii) $r_{mm}$. For the manually annotated radius ($r_{px}$) and the caliber measured radius ($r_{mm}$) uncertainties, we considered, respectively, $\frac{a}{\sqrt{3}}$, with *a* being 1 px (image resolution) and 0.3 mm (calculated through repeated measures on the same berries). The uncertainties of $V_{I,px}$ and $V_{b,mm}$ were computed by applying GUM to Equations (Equation 7) and (Equation 8).
(7)$$V_{I,px}=\sum_{k=1}^{N_{i}}\frac{4}{3}\cdot \pi r_{px}^{3}$$
(8)$$V_{b,mm}=\sum_{k=1}^{N_{T}}\frac{4}{3}\cdot \pi r_{mm}^{3}$$

To compute the uncertainty of ${\tilde{V}}_{b,mm}$, we need to consider following variables: (i) conversion factor $C_{px-mm}$ (and thus the camera–grape distance *d* and focal length *f*), (ii) the average ratio *R*, and (iii) the estimated grape bunch volume ${\tilde{V}}_{I,px}$ expressed in ${\mathrm{px}}^{3}$.

We chose to associate to *d* uncertainty $\sigma_{d}=25$ mm. This assumption was defined because the positioning of the grape bunch with respect to the camera varies depending on where it is located on the vine (the vine distance was fixed to 500 mm from the camera). We empirically defined $\sigma_{d}$ as the half thickness of the vine considered. In real-case applications, this parameter could be set to higher values to account also for the uncertainty on *d*. The focal length *f* (in pixels) was estimated through a standard camera calibration procedure as described in Section 2.3 with uncertainty equal to $\sigma_{f}=2$ px.

The uncertainty of parameter *R* was computed as the standard deviation of the observed ratio between the volume of the entire grape bunch $V_{b,mm}$ and the volume of the visible grapes $V_{I,mm}$, averaged across all the images in the test dataset. This resulted in a value of $\sigma_{R}=0.32$.

The uncertainty of ${\tilde{V}}_{I,px}$ (Equation (Equation 5)) was obtained by computing the RMSE between the estimated visible pixel volumes for each Image I in the test dataset, ${\tilde{V}}_{I,px}$, and their corresponding reference volume $V_{I,px}$ (Equation (Equation 7)). We recall that the main difference between the two volumes stands in the radius used in the formulation, which is the predicted mean berry radius obtained from STEWIE in the case of ${\tilde{V}}_{I,px}$ and the manually annotated radius of each visible berry in the case of $V_{I,px}$.

To compute the uncertainty of the total estimated volume of each bunch *b* (${\tilde{V}}_{b,mm}$), we applied GUM to Equation (Equation 6).

## 3. Results and Discussion

### 3.1. Model Validation

Model performances in berry counting, radius, and volume estimation are shown in Figure 4. For the berry counting task applied to the test dataset (red box plot in Figure 4), the resulting mean error ME is −1.57 with a standard deviation $\sigma_{E}$ equal to 1.9. The mean percentage error MPE is equal to −5% with a standard deviation of 6.3%. For the sake of completeness, we also present the individual count for each image in Table 2. The negative value highlights that STEWIE tends to underestimate the number of berries, probably due to occlusions (e.g., background berries completely or partially hidden by foreground berries).

Table 3 shows the comparison between the validation and test datasets following the procedure in Section 2.4.1 to obtain comparable results between single-cluster-per-image and multiple-clusters-per-image scenarios. To enhance clarity and emphasize distinctions, the results are presented by dividing the outcomes associated with the validation dataset based on the grape variety. Results obtained using the validation dataset are fully comparable with those obtained with the test dataset. The MAE column indicates that counting errors may reach around 32 berries for an image containing an average of 14 clusters. To understand whether this error is acceptable, it is essential to determine the total number of berries in the validation and test datasets. Based on the data shown in Table 1, we hypothesize that the average number of berries for each cluster is 50 (obtained averaging column $N_{T}$ of Table 1). Expanding this value to 14 clusters, we obtain an average number of berries equal to 700. This value is utilized to compute the normalized MAE (column ${\mathrm{MAE}}_{norm}$ in Table 3). The average MAE is equal to 3.1%, with a standard deviation of 0.8%. This number is entirely suitable for the application considered.

Regarding the mean radius estimation task (green box plot in Figure 4), STEWIE achieves a mean estimation error ME of 0.15 px with a standard deviation of 1.5 px, corresponding to a mean percentage error MPE of 0.8% with a standard deviation of 7%, which is a promising result.

Regarding the volume estimation, we first compare the reference visible volume using the manual annotations, $V_{I,px}$, with the estimated visible volume obtained using the mean berry radius predicted by STEWIE, ${\tilde{V}}_{I,px}$ (blue box plot in Figure 4). We obtain a mean estimation error ME of $-20.7\cdot 10^{3}$${\mathrm{px}}^{3}$ with a standard deviation of $2.9\cdot 10^{5}$${\mathrm{px}}^{3}$ and a mean percentage error MPE of 0.36% with a standard deviation of 20.8%. Even if the average error is almost null, results on volume estimation (in pixels) show a high variability. This inconsistency in results suggests that the low bias of the volume estimates may not be reliable.

As described in Section 2.4.3, once the *visible* volume estimation ${\tilde{V}}_{I,px}$ is evaluated, we examine the volume of whole bunches $V_{b,mm}$. Figure 5 shows the estimation error $E_{k}$ obtained as the difference between estimated volume ${\tilde{V}}_{b,mm}$ and nominal volume $V_{b,mm}$ for each image *k* in the test dataset, coupled with the corresponding uncertainty. Uncertainties of each quantity considered are computed as described in Section 2.5. Figure 5 shows that the measured volume of almost every grape bunch is compatible with its nominal value.

For each grape bunch, we compute the coefficients of variability (CoV) (i.e., the ratio between the standard deviation and the mean of each measure). All measurements show high variability: average CoV within 30 measurements equals 24% with a standard deviation of 7%. This effect is partially due to the variability of ${\tilde{V}}_{I,px}$. Other parameters that affect this result are the uncertainty of (i) $C_{px-mm}$ and (ii) *R*. To understand this result, a further analysis of the uncertainty associated with the measurement must be conducted, as described in the following section.

### 3.2. Uncertainty Analysis

As described in Section 2.5, we computed the measurement uncertainty of every parameter that plays a role in the overall computation of the volume’s uncertainty. To compute the uncertainty on the total estimated volume of each bunch *b* (${\tilde{V}}_{b,mm}$), we applied GUM to Equation (Equation 6).

Table 4 shows the summary of the variables coupled with their corresponding uncertainty σ and a brief description of the uncertainty source. The last column shows their average percentage contribution to the overall uncertainty (UPC). From this analysis, it is evident that the most impact on the overall uncertainty was due to ${\tilde{V}}_{I,px}$, which is estimated considering the mean berry radius of the visible berries produced by STEWIE.

The uncertainty on the estimated volume in pixel resulted in $\sigma_{{\tilde{V}}_{I,px}}=2.8\cdot 10^{5}$${\mathrm{px}}^{3}$ and the average uncertainty on the reference volume in pixel results in $\sigma_{V_{I,px}}=1.8\cdot 10^{4}$${\mathrm{px}}^{3}$. We adopted the average uncertainty on the reference because, by applying GUM, we obtained individual uncertainties for each $V_{k,px}$. It is worth noting that the uncertainty on the estimated visible volume, $\sigma_{{\tilde{V}}_{I,px}}$, exceeds by more than an order of magnitude the average uncertainty on its reference, $\sigma_{V_{I,px}}$.

While the uncertainty on the visible volume ($\sigma_{{\tilde{V}}_{I,px}}$) was the most prominent (45%), it can be noted that the uncertainties on the camera–grape cluster distance *d* ($\sigma_{d}$) and on the total/visible volume ratio *R* ($\sigma_{R}$) had a combined impact that was greater than 50%. By adopting strategies to increase the confidence level on *d* and *R*, it could be possible to halve the uncertainty on the visible volume estimation formulation.

For each of the K=30 images of the test dataset (depicting single grape bunches *b*), it is shown in Figure 5 that the uncertainty on the estimated total volume of the bunch ${\tilde{V}}_{b,mm}$ was greater than the reference total volume $V_{b,mm}$ by more than an order of magnitude (considering all bunches, we obtained a mean uncertainty of 42 ${\mathrm{mm}}^{3}$, approximately equal to the 20% of the estimated volume). This result is certainly not satisfactory and could be improved by reducing the uncertainty related to the estimation of the total/visible volume ratio *R* and of the camera–grape cluster distance *d*. To this aim, a possible solution is to adopt depth cameras in a 2D–3D fusion fashion to always know the actual position of the bunch with respect to the camera (*d*) and consequently design a better formulation for parameter *R*. Moreover, the volume was elevated at the power of three, which greatly emphasizes the effect of small errors in the estimation of the average berry radius. Additionally, it is worth mentioning that the RMSE on the volume estimation appeared unbiased (average error close to zero). Thus, averaging multiple bunches could lead to favorable outcomes in whole-orchard analysis.

## 4. Conclusions

In this article, we conducted a metrological validation of the performance of the weakly supervised neural network named STEWIE introduced in our previous work [13], which directly outputs both the number of individual grape berries and their average radius from 2D images. This is a novel feature not yet explored by other works, especially for small fruits such as grape berries. The contribution of this article stands in the thorough validation and uncertainty evaluation of the model’s performance, a topic often overlooked in precision agriculture research.

The validation was conducted on the two outputs of the model: (i) the visible berry counting in the images and (ii) the corresponding berry radius estimation. From these two parameters, it is possible to compute the overall grape bunch volume, which is the key information needed by vinegrowers to accurately estimate the yield. To assess which parameter contributes to the most uncertainty in the final volume estimation, we applied the GUM and derived the UPC of each parameter. This analysis highlighted that half of the total uncertainty on the volume is due to the camera–object distance *d* and parameter *R* used to take into account the proportion of visible grapes with respect to the total grapes in the grape cluster. As a result, by using more reliable sensors to measure *d* such as depth cameras, our model performance improves.

Finally, since winegrowers are more interested in the whole orchard yield volume information while taking into account the uncertainty of the measurement at the same time, we aim to further improve STEWIE model and incorporate the uncertainty estimation on the final volume output in its design. The complete system will be developed and deployed on a robust embedded device able to acquire every information needed coupled with the corresponding frames so that the whole orchard can be analyzed effortlessly.

## Figures and Tables

**Figure 1 sensors-24-07305-f001:**
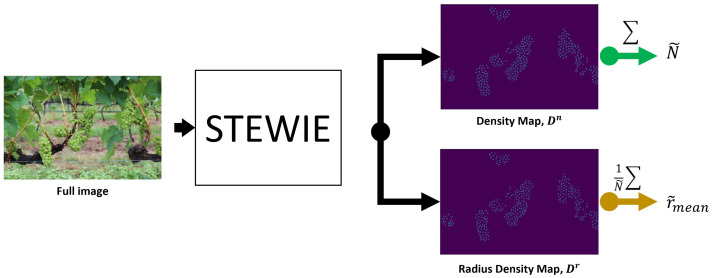
Scheme of the inference process. The image is elaborated by the custom neural network and two probability density maps are returned as output. Pixel densities are summed to compute the estimate of the number of berries $\tilde{N}$ and their average size ${\tilde{r}}_{\mathrm{mean}}$.

**Figure 2 sensors-24-07305-f002:**
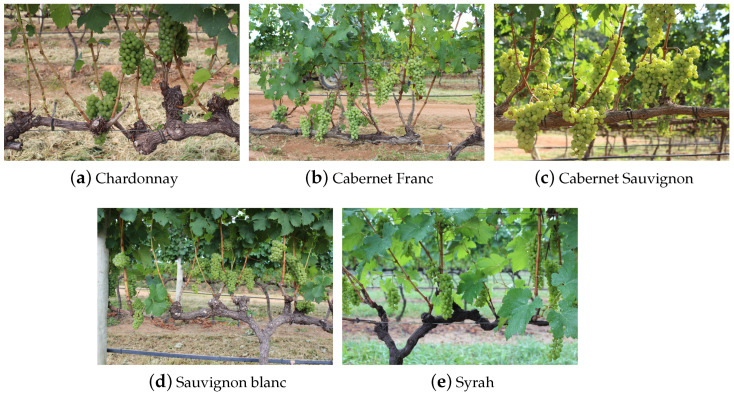
Image examples taken from the Embrapa WGISD dataset [14].

**Figure 3 sensors-24-07305-f003:**
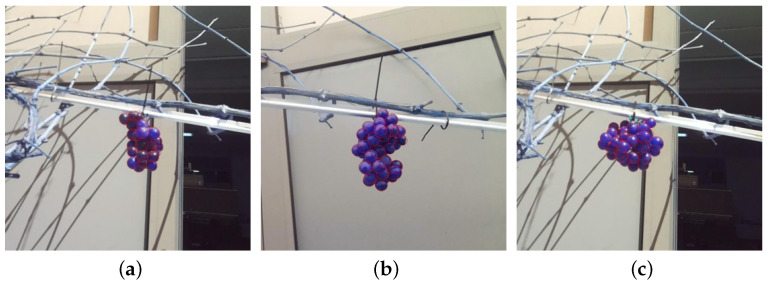
(**a**–**c**) Image examples of the test dataset along with manual annotations overlaid.

**Figure 4 sensors-24-07305-f004:**
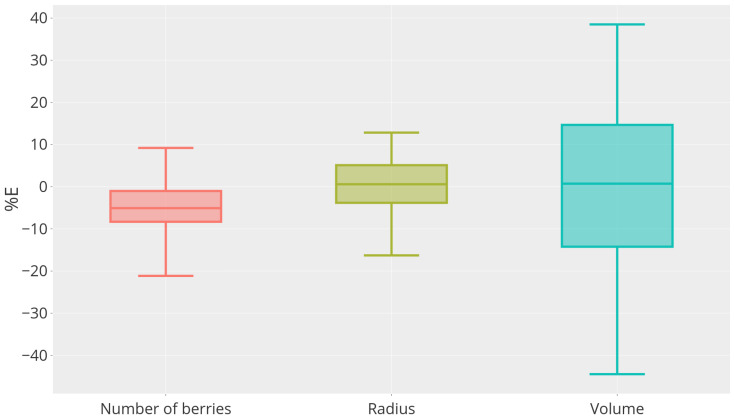
Boxplot of the relative error $RE_{k}$ in % computed for the number of visible berries, the value of the average radius, and the visible volume of the bunch depicted in the test images.

**Figure 5 sensors-24-07305-f005:**
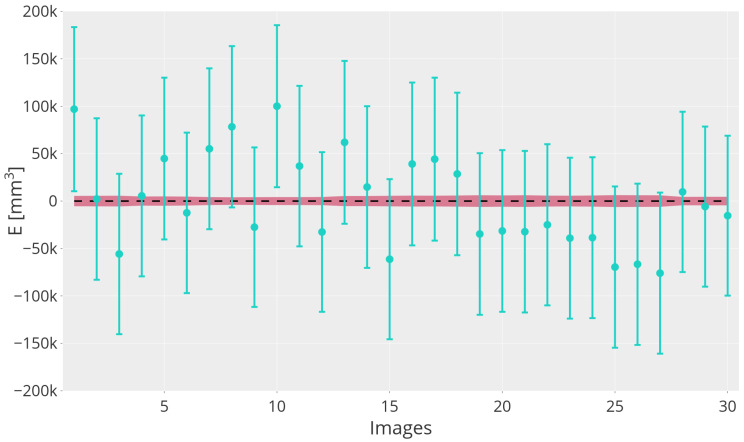
Total bunch volume error $E_{k}$ (difference between estimated volume ${\tilde{V}}_{b,mm}$ and nominal volume $V_{b,mm}$) for each image *k* in the test dataset, coupled with the corresponding uncertainty. The shaded red area corresponds to the 95% confidence interval of the ground truth.

**Table 1 sensors-24-07305-t001:** Summary of the validation data per bunch (ground truth), including (i) number of total berries ($N_{T}$), (ii) number of visible berries in each of the three images ($0^{\circ }$, $120^{\circ }$, and $240^{\circ }$ and $N_{1}$, $N_{2}$ and $N_{3}$, respectively), (iii) average radius of the bunch ($r_{mean,T}$, expressed in millimeters), and (iv) average radius of the visible berries in each image ($r_{mean,1}$, $r_{mean,2}$ and $r_{mean,2}$, all expressed in pixels).

Bunch ID	$N_{T}$	$N_{1}$	$N_{2}$	$N_{3}$	$r_{\mathrm{mean},T}\;\pm \;\sigma$	$r_{\mathrm{mean},1}$	$r_{\mathrm{mean},2}$	$r_{\mathrm{mean},3}$
Bunch_01	47	30	28	26	10.1±0.35 mm	25 px	23 px	21 px
Bunch_02	52	37	32	41	9.3±0.52 mm	21 px	21 px	19 px
Bunch_03	36	21	25	24	9.3±0.44 mm	24 px	23 px	19 px
Bunch_04	39	23	23	21	9.2±0.59 mm	25 px	22 px	19 px
Bunch_05	55	31	31	27	9.5±0.57 mm	24 px	23 px	22 px
Bunch_06	51	32	35	29	10.1±0.36 mm	23 px	22 px	25 px
Bunch_07	63	33	39	34	9.8±0.42 mm	22 px	21 px	22 px
Bunch_08	52	30	34	29	10.1±0.45 mm	22 px	21 px	21 px
Bunch_09	76	41	43	48	9.5±0.40 mm	20 px	20 px	19 px
Bunch_10	40	29	24	24	9.5±0.38 mm	22 px	21 px	20 px

**Table 2 sensors-24-07305-t002:** Estimated number of berries for each of the three images of the 10 bunches comprising the test dataset. Inside the parentheses, the difference with respect to the ground truth (refer to Table 1 for comparison).

Bunch ID	${\tilde{N}}_{1}$	${\tilde{N}}_{2}$	${\tilde{N}}_{3}$
Bunch_01	30(0)	28(0)	26(0)
Bunch_02	32(−5)	35(+3)	38(−3)
Bunch_03	20(−1)	24(−1)	23(−1)
Bunch_04	22(−1)	23(0)	20(−1)
Bunch_05	28(−3)	28(−3)	21(−6)
Bunch_06	33(+1)	37(+2)	36(−3)
Bunch_07	31(−2)	36(−3)	31(−3)
Bunch_08	28(−2)	30(+1)	29(0)
Bunch_09	38(−3)	42(−1)	45(−3)
Bunch_10	24(−5)	23(−1)	24(0)

**Table 3 sensors-24-07305-t003:** Evaluation metrics of the berry counting task computed for both test and validation images, divided by grape variety. $I_{T}$ refers to the number of images of the variety in the dataset. The values corresponding to the test dataset are the estimated values derived from the analysis detailed in Section 2.4.1. Column ${\mathrm{MAE}}_{norm}$ is computed considering an average number of berries in each image equal to 700.

Variety	Dataset Used	ME	MAE	${\mathrm{MAE}}_{\mathrm{norm}}$	RMSE	$I_{T}$
Chardonnay	Validation	−26.8	32.2	4.5%	45.5	13
Cabernet Franc	Validation	−0.7	17.0	2.4%	20.5	22
Cabernet Sauvignon	Validation	4.0	21.1	2.9%	30.7	14
Sauvignon Blanc	Validation	5.7	23.8	3.3%	31.6	15
Syrah	Validation	−8.6	16.5	2.3%	21.7	11
Flame	Test	−21.8	21.9	3.1%	23.1	30

**Table 4 sensors-24-07305-t004:** Sources of uncertainty definition. To each variable, the corresponding uncertainty and UPC are shown for quick reference.

Variables	Uncertainty	Definition and Reason of Uncertainty	UPC
*d*	25 mm	Uncertainty set considering the vine thickness and the eventual cluster misplacement that could modify the default value of *d*.	22.1%
*f*	2 px	Uncertainty depends on the quality of the images and of the pattern used for the camera calibration procedure.	0.3%
*R*	0.32	Uncertainty set as the standard deviation of the values that were averaged to compute *R* (e.g., the ratios between the visible and the total volume of the bunches for each photo).	32.6%
${\tilde{V}}_{I,px}$	$2.8\cdot 10^{5}$ ${\mathrm{px}}^{3}$	Uncertainty set as the RMSE between the estimated pixel volumes for each Image I in the test dataset, ${\tilde{V}}_{I,px}$, and their corresponding reference volume $V_{I,px}$.	45%

## Data Availability

Data are contained within the article.

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
