# Peer review of "A Stride Toward Wine Yield Estimation from Images: Metrological Validation of Grape Berry Number, Radius, and Volume Estimation"

_sensors, 2024, doi:10.3390/s24227305_

Round 1
Reviewer 1 Report
Comments and Suggestions for Authors
The whole paper does not design its own network or method, but just cites the content in reference 21. I recommend designing your own approach and addressing key scientific questions. It is suggested to rethink and design the whole paper structure.
Author Response
Q1: The whole paper does not design its own network or method, but just cites the content in reference 21. I recommend designing your own approach and addressing key scientific questions. It is suggested to rethink and design the whole paper structure.
A1: We thank the reviewer for his/her valuable comment. We agree that the structure of the paper should be rethinked and adjusted also according to other reviewers’ comments. We hope that the new submission better adheres to the reviewers’ expectations. We would also like to stress that the submitted paper is a technical extension of the preliminary work presented in reference 21 (now 13) that better explains the methodology of the network (which is novel, albeit being shortly presented in reference 13) and showing a validation procedure, not addressed in reference 13. To emphasize these two aspects that differentiate the present submission from reference 13 (previous work), we added a few lines in the introductory section.
Reviewer 2 Report
Comments and Suggestions for Authors
1.The abstract mainly reflects the research background, general content, and results, without focusing on the manifestation of innovation. Therefore, the completeness of the abstract is insufficient.
2.The author took about 3 pages to complete the introduction, which the reviewer believes is not conducive to reflecting the core issue.
3.Many formulas are closely followed by words such as' where ', which essentially belong to the same paragraph as the previous content. Please correct the entire text according to the specific situation.
4.The data in Table 5 has a high degree of dispersion, which is related to specific grape varieties. Does this contradict the applicability of the research results?
5.In the conclusion section, please do not elaborate on the research background and necessity, but directly introduce your own research results.
6.The reviewer believes that the conclusion section does not highlight specific innovative performance.
7.The full text description is not concise, and the reviewer believes that multiple parts can be further simplified and there is no need to use too much description.
Comments on the Quality of English LanguageOverall, English writing is fluent, but the abstract and conclusion are relatively simple and cannot fully demonstrate innovation; Many specific defect pages within the manuscript need to be corrected one by one.
Author Response
Q1: The abstract mainly reflects the research background, general content, and results, without focusing on the manifestation of innovation. Therefore, the completeness of the abstract is insufficient.
A1: We thank the reviewer for this comment. We modified the abstract as suggested to better highlight the novelty of the work.
Q2: The author took about 3 pages to complete the introduction, which the reviewer believes is not conducive to reflecting the core issue.
A2: The instruction paragraph contains both information about the context of the presented work as well as background literature references. However, we do agree with the reviewer that the paragraph can be shortened and rewritten to better highlight the focus of the presented work. As a result, we modified the paragraph with this idea in mind, hoping to meet the reviewer’s expectations.
Q3: Many formulas are closely followed by words such as' where ', which essentially belong to the same paragraph as the previous content. Please correct the entire text according to the specific situation.
A3: We agree that there are necessary improvements that should be made to the mathematical parts to improve clarity. We modified these parts accordingly.
Q4: The data in Table 5 has a high degree of dispersion, which is related to specific grape varieties. Does this contradict the applicability of the research results?
A4: Thanks for the comment. Indeed, this part was not clear. We modified Table 5 adding a column where MAE normalized (MAE_norm) has been inserted. MAE_norm shows the ratio between MAE and the mean number of berries in a single image considering 14 bunch per image. The MAE_norm values show that the counting error is acceptable for this kind of applications. We modified the text accordingly to explain this part.
Q5: In the conclusion section, please do not elaborate on the research background and necessity, but directly introduce your own research results.
A5: We thank the reviewer for this suggestion. We merged the discussion and conclusion sections into one that better adheres with traditional conclusive paragraphs that summarize the whole paper and the results.
Q6: The reviewer believes that the conclusion section does not highlight specific innovative performance.
A6: Referring to the previous answer, we merged the discussion and conclusion paragraphs into one that better highlights the scientific advancements presented in the paper.
Q7: The full text description is not concise, and the reviewer believes that multiple parts can be further simplified and there is no need to use too much description.
A7: We agree with the reviewer on this aspect. We carefully revised the text and removed unnecessary descriptive portions.
Q8: Overall, English writing is fluent, but the abstract and conclusion are relatively simple and cannot fully demonstrate innovation; Many specific defect pages within the manuscript need to be corrected one by one.
A8: We carefully revied the manuscript with the reviewer’s comments in mind, simplifying the whole paper and removing unnecessary portions, better focusing on the specific technical advancements with respect to our previous work (ref 13) and the literature.
Reviewer 3 Report
Comments and Suggestions for Authors
The article under consideration is devoted to the assessment of grape yields. The number, radius and volume of berries were estimated. This assessment is important for wine producers. In my opinion, the study was carried out at a high level, the methods are modern and correspond to the task. The conclusions correspond to the results obtained. The work is of practical importance for wine producers. The illustrations are made at a high level. In my opinion, the article can be published without changes.
Author Response
Q1: The article under consideration is devoted to the assessment of grape yields. The number, radius and volume of berries were estimated. This assessment is important for wine producers. In my opinion, the study was carried out at a high level, the methods are modern and correspond to the task. The conclusions correspond to the results obtained. The work is of practical importance for wine producers. The illustrations are made at a high level. In my opinion, the article can be published without changes.
A1: We thank the reviewer for his/her evaluation.
Reviewer 4 Report
Comments and Suggestions for Authors
The manuscript sensors-3274792 entitled "A stride toward wine yield estimation from images: grape berry number, radius, and volume estimation through density maps" submitted by Lanza et al. presents an interesting research activity regarding grape yield estimation from 2d colour images.
Considering the topic investigated in the present manuscript and the main aims and scopes of the journal sensor I belive that this article may deserve to be received in the sensor journal and may be of interest to its readers.
The first aspect related to the evaluation of this article is its structure. It, in fact, does not have the normal structure sanctioned by the instructions for authors but follows a structure of its own. This makes it difficult to lacture and understand the experimental activities conducted. Therefore, I urge the authors to reformulate the manuscript in the structure that includes Introduction, Materials and Methods, Discussion and Conclusion.
Anyway, the manuscript is of good quality and presents very interesting data. The Introduction part well presents the state of the art while the materials and methods are very thorough. The results, are quite concise but fine. The Discussion is just a repetition and detailing of the results and should be joined to those. The conclusion presents a brief discussion of the results; this part should be renamed discussion and should be expanded with other speculations and comparison with similar research activities. The actual conclusions should present only concluding remarks on what was observed in the experiment.
My specific comments are detailed in the attached pdf file.

Author Response
Q1: The first aspect related to the evaluation of this article is its structure. It, in fact, does not have the normal structure sanctioned by the instructions for authors but follows a structure of its own. This makes it difficult to lacture and understand the experimental activities conducted. Therefore, I urge the authors to reformulate the manuscript in the structure that includes Introduction, Materials and Methods, Discussion and Conclusion.
A1: We thank the reviewer for this comment. We reviewed the manuscript structure aligning it with the format requested in the journal guidelines.
Q2: The conclusion presents a brief discussion of the results; this part should be renamed discussion and should be expanded with other speculations and comparison with similar research activities. The actual conclusions should present only concluding remarks on what was observed in the experiment.
A2: Thanks for the comment. As in the previous answer, we reviewed the manuscript structure aligning it with the format requested in the journal guidelines.
Q3: My specific comments are detailed in the attached pdf file
A3. We addressed all the comments listed in the pdf file. In particular, we correct the typos requested and we changed the citation layout.
Round 2
Reviewer 1 Report
Comments and Suggestions for Authors The manuscript has been sufficiently improved to warrant publication in Sensors.Reviewer 4 Report
Comments and Suggestions for Authors
The manuscript was reviewed according to my comments improving its quality.